# A Novel In Vivo Active Pemphigus Model Targeting Desmoglein1 and Desmoglein3: A Tool Representing All Pemphigus Variants

**DOI:** 10.3390/biology12050702

**Published:** 2023-05-11

**Authors:** Roberta Lotti, Claudio Giacinto Atene, Emma Dorotea Zanfi, Matteo Bertesi, Carlo Pincelli, Tommaso Zanocco-Marani

**Affiliations:** 1DermoLAB, Department of Surgical, Medical, Dental and Morphological Sciences, University of Modena and Reggio Emilia, 41124 Modena, Italy; 2Hematology Section, Department of Medical and Surgical Sciences, University of Modena and Reggio Emilia, 41124 Modena, Italy; 3Department of Life Sciences, University of Modena and Reggio Emilia, 41125 Modena, Italy

**Keywords:** autoimmune disease, skin models, pemphigus, desmoglein

## Abstract

**Simple Summary:**

Pemphigus is a severe and heterogeneous autoimmune disease determining the formation of blisters in the skin and mucosae. Life quality of patients is indeed deteriorated by the disease. The different kinds of pemphigus are caused by the presence of autoantibodies targeting different autoantigens. Many of these are proteins belonging to the Cadherin family that physiologically plays a role in the integrity of skin and mucosae. So far, therapies effectively curing the disease have not been developed and actual strategies are mainly aimed at the control of symptoms. In order to develop specific therapies, efficient disease models are much needed. The most common forms of pemphigus are those characterized by the presence of autoantibodies against the cadherin DSG3 (affecting mainly the mucosae), the cadherin DSG1 (affecting mainly the epidermis), or both simultaneously (mucocutaneous). Here, we present a mouse model where animals can develop the three types of disease. We believe this model, encompassing the three main forms of pemphigus, represents a very robust and needed proving ground for the development of new therapies.

**Abstract:**

**Background:** Pemphigus is a life-threatening blistering autoimmune disease. Several forms, characterized by the presence of autoantibodies against different autoantigens, have been described. In Pemphigus Vulgaris (PV), autoantibodies target the cadherin Desmoglein 3 (DSG3), while in Pemphigus foliaceous (PF) autoantibodies target the cadherin Desmoglein 1 (DSG1). Another variant, mucocutaneous Pemphigus, is characterized by the presence of IgG against both DSG1 and DSG3. Moreover, other forms of Pemphigus characterized by the presence of autoantibodies against other autoantigens have been described. With regard to animal models, one can distinguish between passive models, where pathological IgG are transferred into neonatal mice, and active models, where B cells deriving from animals immunized against a specific autoantigen are transferred into immunodeficient mice that develop the disease. Active models recreate PV and a form of Pemphigus characterized by the presence of IgG against the cadherin Desmocollin 3 (DSC3). Further approaches allow to collect sera or B/T cells from mice immunized against a specific antigen to evaluate the mechanisms underlying the onset of the disease. **Objective:** To develop and characterize a new active model of Pemphigus where mice express auto antibodies against either DSG1 alone, or DSG1 and DSG3, thereby recapitulating PF and mucocutaneous Pemphigus, respectively. In addition to the existing models, with the active models reported in this work, it will be possible to recapitulate and mimic the main forms of pemphigus in adult mice, thus allowing a better understanding of the disease in the long term, including the benefit/risk ratio of new therapies. **Results:** The new DSG1 and the DSG1/DSG3 mixed models were developed as proposed. Immunized animals, and subsequently, animals that received splenocytes from the immunized donors produce a high concentration of circulating antibodies against the specific antigens. The severity of the disease was assessed by evaluating the PV score, evidencing that the DSG1/DSG3 mixed model exhibits the most severe symptoms among those analyzed. Alopecia, erosions, and blistering were observed in the skin of DSG1, DSG3 and DSG1/DSG3 models, while lesions in the mucosa were observed only in DSG3 and DSG1/DSG3 animals. The effectiveness of the corticosteroid Methyl-Prednisolone was evaluated in the DSG1 and DSG1/DSG3 models, that showed only partial responsiveness.

## 1. Introduction

Pemphigus is a complex and heterogeneous autoimmune cutaneous disease. The most frequent form is Pemphigus Vulgaris (PV), mediated by anti-Desmoglein 3 (DSG3) autoantibodies, where blisters occur mainly in the mucous membrane, whereas Pemphigus Foliaceus (PF) is instead caused by anti-Desmoglein 1 (DSG1) IgG with blisters occurring mainly in the epidermis. Mucocutaneous Pemphigus is determined by the presence of autoantibodies targeting both DSG3 and DSG1 and lesions in both sites. Other atypical forms of Pemphigus have been described and are distinguished by the presence of autoantibodies aimed at also other autoantigens, such as the cadherin Desmocollin 3 (DSC3) and others [1]. Different models for the study of Pemphigus have been established: from in vitro assays and ex vivo organ culture, to more elaborated in vivo models [2]. There are two main approaches to generate an animal model of Pemphigus: the passive transfer of pathogenic IgG to neonatal and adult mice [3,4], or the transfer of immune cells actively producing IgG to immunodeficient mice [5]. To obtain an active model for PV, a DSG3-/- mouse can be immunized against the knock-out protein by inoculating murine recombinant DSG3 (rDSG3) to induce production of auto-reactive antibodies [5]. Mice should be immunized with several doses of recombinant protein to mount an appropriate immune response. Then, splenocytes are collected from animals and transferred into immunodeficient Rag2-/- recipient mice through tail vein injection; in the host Rag2-/- population a fraction of the splenocytes will produce antibodies against endogenously expressed DSG3 [6]. The active DSG3 pemphigus mouse model was successfully used in many studies. For instance, since among splenocytes, both T and B cells are represented and the model has contributed to enlighten the role of T cells in PV. In fact, it was demonstrated that both B and T cells are involved in the loss of tolerance against DSG3 [7], and that polyclonal B cells can be committed to pathogenic IgG production by a single DSG3-reactive T cells clone. [8,9]. A similar approach was used to develop an active model for Atypical Pemphigus, where the disease is sustained by the presence of IgG against autoantigens different from DSG1 and DSG3. This latter version of the active model was obtained by inducing the expression of autoantibodies against Desmocollin 3 (DSC3). For this model to obtain splenocytes reactive against DSC3, WT mice were repeatedly inoculated with rDSC3 to break the immune tolerance to DSC3 and produce pathological antibodies [10], since DSC3 knockout does not allow neonatal survival. These models provided a useful platform to follow the progression of the disease in animals and to assess the efficacy of experimental drugs. The active models described were further developed. For instance, sera and T cells were collected from HLA-restricted mice immunized against DSG3 [11] or DSC3 [12] to obtain information on the mechanisms underlying the onset of the disease regarding the role of pathogenetic antibodies, HLA (Human Leukocyte antigen) and T cells.

At the current state of knowledge, in vivo active mouse models are one of the most valuable solutions to recapitulate the cellular complexity of Pemphigus disease seen in patients, and a method to evaluate long-term benefits and potential side-effects of experimental therapies on various tissues and functions. On the one hand, the availability of active mouse models of PV and Atypical Pemphigus (DSC3) responds to those research needs, while on the other hand, no active mouse model of PF (DSG1) or Mucocutaneous Pemphigus, where autoantibodies target both DSG1 and DSG3, has been generated yet. In the present work, we have developed and validated two novel active mouse models representing PF and Mucocutaneous Pemphigus.

## 2. Materials and Methods

### 2.1. Recombinant Protein Expression and Purification

Mouse Desmoglein 1 (mDSG1) was expressed as the 6xHis-tag fusion protein in Sf9 insect cells as follows. Briefly, the total cDNA was obtained by retro-transcription performed on mouse skin that derived the total RNA. The DNA sequence encoding the extracellular portion of mDSG1 was then amplified by PCR using the specific primers: FW 5′-CTTTTATGGAATGGATCAAGTTTGCTGCAG-3′ and RV 5′-AGGGACCGAAGTGAACGTGTCTC-3′. The PCR product was subcloned in the pFastBac/C-His TOPO vector (Bac-to-Bac C-His TOPO^®^ cloning and expression Kit, Invitrogen, Carlsbad, CA, USA) carrying a C-terminal His-tag, as analogously reported [10]. Sf9 cells were then transfected with the recombinant DNA using Cellfectine II reagent (Invitrogen, Carlsbad, CA, USA) by the manufacturer’s guidelines. Following three rounds of infection, we collected the viral supernatant (P3). This concentrated baculoviral stock was used to infect Sf9 cells again for protein production. As previously reported, the P3 baculoviral stock for rDSG3 production was a kind gift from Dr. Amagai [5]. For production, 2–4 × 10^6^/mL Sf9 cells were infected with concentrated P3 recombinant baculoviral stock at MOI 5, as previously reported [10]. Recombinant proteins were then purified as previously detailed in [10].

### 2.2. Mice Immunization

B6;129X1-Dsg3tm1Stan/J and B6(Cg)-Rag2tm1.1Cgn/J adult mice were obtained from Charles River Italia (Calco, Italy) and maintained at the Animal Facility of the University of Modena and Reggio Emilia. Ten weeks-old Dsg3-/- mice were immunized as described in [10]. The immunization protocol leading to the loss of tolerance to DSG1 was performed in 7-weeks-old WT mice according to our previous protocol that was set up for DSC3 [10], which we adapted from the one described by Hirose et al. [13]. Control animals (CNTRL) were obtained by immunization with a sample derived from non-infected Sf9 cells that underwent the same procedure as infected cells, according to the tolerance-breaking protocol. To verify the production of antibodies against DSG1 and DSG3, indirect immunofluorescence was performed on WT healthy skin. Briefly, murine healthy skin slides were incubated with 1:50-diluted serum from immunized animals. Subsequently, presence of antibodies was detected with a secondary Alexa Fluor 488 anti-mouse antibody (Invitrogen Corporation, Carlsbad, CA, USA).

### 2.3. Ethics Statement

Animal studies and animal procedures were first approved by the Animal Welfare Committee of the University of Modena and Reggio Emilia, and then by the Italian Ministry of Health (406/2015-PR). Protocol was carried out in accordance with the Italian Institute of Health guidelines.

### 2.4. Adoptive Transfer of Splenocytes and Treatments

Splenocytes were isolated from immunized mice and pooled from two or more immunized mice. 20 × 10^6^ DSG1 or DSG3-reactive splenocytes were transferred via tail vein injection in 10–11-weeks-old Rag2-/- mice. To develop the DSG1/DSG3 mouse model, 10 × 10^6^ splenocytes per each immunization type were injected per Rag2 mouse. Methyl-prednisolone (m-PSL) (Solu-Medrol^®^ Pfizer, Tokyo, Japan) was provided intraperitoneally at a dose of 100 mg/kg daily, starting from day 7 after the adoptive transfer and continued for 4 weeks, till day 35, in DSG1 and DSG1/DSG3 treated mice. PBS was used as a negative control (CNTRL).

### 2.5. PV Score

To monitor the development of the disease in the pemphigus mouse models, PV score was evaluated weekly according to Table 1. Erosions and erythema were scored 1 for every occurrence in different areas of the body as described in the table. Alopecia was scored 1 in the face and back, and 0.5 in the neck and abdomen. Additionally, redness and erythema were evaluated with a score from 0.5 to 1, depending on their extension and intensity.

### 2.6. Mouse DSG1 and DSG3 ELISA Assay

ELISA to determine titers of anti-DSG1 and anti-DSG3 autoantobodies was run according to Lotti et al [9]. Briefly, 96-well plates were coated with purified recombinant murine rDSG1 or rDSG3 (50 μL of 4 μg/mL) at 4 °C overnight. After several washes with washing buffer (PBS/0.1% Tween), plates were incubated in blocking buffer (wash buffer + 5% BSA) and conserved at 4 °C at least overnight. Sera were 200-fold diluted in assay diluent and incubated for 1 h at room temperature on coated plates. After several washes, plates were then incubated with a HRP-conjugated goat polyclonal secondary anti-mouse IgG antibody (AbCam, Cambridge, UK) for at least 1 h at room temperature. After washes, the TMB substrate solution was added. The reaction was stopped after 30 min by adding stop solution. Plates were read at 450 nm with reference at 620 nm by iMark™ Microplate Absorbance Reader (BioRad, Hercules, CA, USA). Sera from either DSG1- or DSG3-immunized mice were used as positive controls, while serum from non-treated WT mice was used as a negative control. The index value was defined by the formula: $$Index=\frac{OD\;sample-OD\;negative\;control}{OD\;positive\;control-OD\;negative\;control}\times 100$$

### 2.7. Statistical Analysis

Data are presented as mean ± SEM and analyzed by Prism Software (Graph Pad Software V9.0, San Diego, CA, USA). Briefly, a two-tailed unpaired Student’s *t*-test was used for statistical comparisons between two groups, while one-way ANOVA or 2-way ANOVA was used for multiple groups, together with an appropriate multiple comparisons test. A *p* value < 0.05 or less was assumed to indicate a statistically significant difference in the compared parameters.

## 3. Results

### 3.1. Development of an Active Model Representing PF and Mucocutaneous Pemphigus: Evaluation of Antibodies Production in Immunized Animals

To develop active models mimicking PF and Mucocutaneous Pemphigus, the extracellular domains of murine DSG3 and DSG1, generally targeted by pathological antibodies, were cloned. Both recombinant proteins (rDSG3 and rDSG1) were produced in insect cells infected by recombinant baculovirus as previously described [10]. In particular, rDSG1 corresponds to the amino-terminal portion of the protein, representing extracellular and Cadherin domains. rDSG3 was used to immunize DSG3-/- mice, while rDSG1 was repeatedly injected in WT mice to break immunological tolerance against DSG1 similarly to [10]. Subsequently, splenocytes were extracted from rDSG3-immunized DSG3-/- mice and from rDSG1-immunized WT animals. To obtain the DSG1 (PF) and the DSG3 (PV) single antigen models, 20 × 10^6^ splenocytes were injected in Rag2-/- mice via the tail vein, while to develop the mixed antigen model of Mucocutaneous Pemphigus (DSG3/DSG1), 10 × 10^6^for each type of reactive splenocytes were i.v. injected in Rag2-/- animals.

To verify that the immunization process occurred correctly, prior to the splenocytes transfer, sera were collected from immunized animals and the presence of antibodies against rDSG1 and rDSG3, respectively, was evaluated by ELISA and immunofluorescence (Figure 1A–C). Figure 1A,B shows the results of an ELISA assay detecting the presence of antibodies against rDSG1 and rDSG3 in immunized animals compared to those untreated (Dil) or treated with adjuvant only (ADJ). We further confirmed the presence of auto-antibodies, anti-DSG1 and anti-DSG3 by indirect immunofluorescence on WT healthy skin (Figure 1C).

### 3.2. Evaluation of the Disease Phenotype of the New Active Models

Rag2-/- mice that received splenocytes isolated from WT animals immunized with rDSG1 spontaneously developed a disease phenotype, characterized by few erosions and a high extent of alopecia and erythema, evaluated by the PV score. Rag2-/- mice that received splenocytes from WT mice immunized with rDSG1, in association with splenocytes from DSG3-/- mice immunized with rDSG3, developed a more acute and intense phenotype with highest PV score (Figure 2A). In particular, the single DSG1 model showed a mild phenotype, compared to both the DSG3 model and the combined DSG1/DSG3 model, the latter showing the more severe disease. Converting the PV score overtime in AUC, the comparisons between the phenotypes of the different models are statistically significant (Figure 2B). The presence of specific anti-DSG1 and anti-DSG3 antibodies was detected in the different models with an increase overtime of the antibody titers, as shown by ELISA (Figure 2C,D). 

Histologic examination of the skin showed a substantial intraepithelial loss of cell-cell adhesion, right above the basal layers of the epidermis, in mice producing antibodies against both DSG3 and DSG1, while loss of adhesion involved the more superficial layers of the epithelium in the DSG1 single model. This difference between the two models was observed also in the bulge of hair follicles (Figure 2E). Since the presence of antibodies against DSG1 or DSG3 might differentially affect the mucosal compartment, we analyzed the oral mucosa of the mixed model animals and of the DSG1 and DSG3 single models. The acantholytic lesions are not evident in the control and DSG1 model, while they are present both in the DSG1/DSG3 mixed model and in the DSG3 single model (Figure 2F).

### 3.3. Validation of the New Models through Methyl-Prednisolone Treatment

To further characterize the two novel DSG1 and DSG1/DSG3 active models, animals were treated with methyl-Prednisolone (m-PSL), a glucocorticoid currently used in the therapy of pemphigus, that was previously used to validate earlier active models. A total of 100 mg/Kg m-PSL was injected i.p. into Rag2-/- mice. Mice were treated daily, starting on day 7 following adoptive transfer of splenocytes. The therapy was maintained for 4 weeks and suspended during the last 28 days of follow up. In the DSG1 model, the effect of m-PSL was statistically significant, although the difference in PV score between treated and untreated groups was limited due to the mild phenotype of the disease (Figure 3A). On the other hand, m-PSL exerted a stronger effect by significantly reducing the PV score in the DSG1/DSG3 mixed model (Figure 3C). As for all these active models, it seems that a more severe phenotype allows a better evaluation of the efficacy of known and experimental therapies. We also evaluated the auto-antibodies titer in m-PSL treated animals compared to CNTRL in both models by ELISA (Figure 3B,D). Blood was collected at days 14 and 28 after splenocytes injection. Results show a decrease in both anti-DSG1 and anti DSG3 antibodies.

## 4. Discussion

With the generation of these new active models, the main forms of pemphigus, as observed in humans, can be reproduced. It is now possible to recapitulate the active models of PV [5], PF, Mucocutaneous Pemphigus and the Atypical anti-DSC3 pemphigus [10]. Active models represent a good option to study pemphigus in vivo for many reasons. They can be extremely helpful to dissect the processes underlying the pathogenesis of pemphigus. For instance, active models might allow a better insight into the role not only of B-cells, but also of T-cells; the activity of the latter being more and more relevant in the disease [7,8,11,12]. 

Several signaling pathways have been implicated in the downstream effects of DSG1 and DSG3 depletion in pemphigus. One of the key pathways is mediated by Rho GTPase signaling. The internalization of DSG1 and DSG3 activates RhoA and Rac1 GTPases, which regulate actin cytoskeleton mechanisms and are involved in the maintenance of cell–cell adhesion. The activation of RhoA and Rac1 in response to DSG1 and DSG3 depletion contributes to the reorganization of the actin cytoskeleton and the breakdown of cell–cell adhesion, which leads to the formation of blisters [14,15,16]. In addition to the Rho GTPase pathway, other signaling pathways have also been implicated in pemphigus. For example, the depletion of DSG1 and DSG3 has been shown to activate the mitogen-activated protein kinase (MAPK) pathway, which regulates various cellular processes, including cell proliferation, differentiation, and apoptosis. The activation of the MAPK pathway in response to DSG1 and DSG3 depletion contributes to the induction of cell death and the formation of blisters [17]. Furthermore, the depletion of DSG1 and DSG3 has also been shown to modulate the activity of other signaling pathways, including protein kinase C (PKC), Rous sarcoma-related kinases (Src), phospholipase C (PLC), and Epidermal growth factor receptor (EGFR), thereby altering adhesive interactions [18]. Overall, the depletion of DSG1 and DSG3 in pemphigus activates multiple signaling pathways, which contribute to the pathogenesis of this autoimmune disease. Understanding the molecular mechanisms underlying these signaling pathways may lead to the development of novel targeted therapies for pemphigus. The DSG1/DSG3 single and mixed animal models might represent a useful tool to verify whether anti DSG1 and anti DSG3 antibodies might show specific activities in the activation of downstream signaling pathways.

In the adult models, one can evaluate novel therapies, allowing for a longer follow up. In addition, the same platform could be used in order to widen the range of autoantigens tested to obtain new insight on the mechanisms underlying less common forms of pemphigus. On a wider horizon, they might be used as a template to establish animal models for other autoimmune diseases. On the other hand, it remains to be elucidated whether in the “mixed antigens” (DSG1/DSG3) active model, an optimal primed T/B cell ratio exists for each antigen in the transferred splenocyte pool to avoid variability in clinical phenotypes. Related to this, we also observed that a more severe disease phenotype allows to better characterize the effectiveness of experimental therapies. Finally, it should be emphasized that active models are time-consuming and have a complex setup.

## 5. Conclusions

In vivo models are indeed demanding. Nevertheless, when compared to other models, such as the passive transfer of pathological IgG to animals that is characterized by the short follow up and the impossibility for the animals to produce their own autoantibodies, they allow to overcome these flaws, since animals are engrafted with splenocytes allowing autoantibodies production and a prolonged follow up. Among the matters to be considered when trying to set up and compare different active models, the number of splenocytes transferred and the method used to immunize the animals (immunization of antigen null mice or tolerance breakage) must be taken into consideration in order to obtain animals developing comparable diseases regarding the severity of the phenotype. Active models have indeed a significant potential owing to their plasticity; in fact, it would be interesting to further broaden this approach to other autoantigens, in order to eventually clarify their role in pemphigus.

## Figures and Tables

**Figure 1 biology-12-00702-f001:**
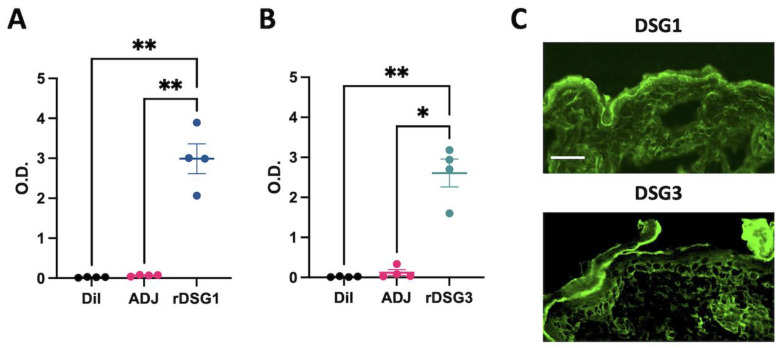
**Evaluation of auto-antibodies against DSG1 and DSG3 in immunized mice.** (**A**) Evauation of the presence of anti-DSG1 Abs in serum samples from WT non-treated mice (Dil), WT Adjuvant-treated animals (ADJ) and rDSG1-immunized animals (rDSG1). (**B**) Evauation of the presence of anti-DSG3 Abs in serum samples from DSG3-/- untreated mice (Dil), DSG3-/- Adjuvant-treated animals (ADJ) and rDSG3-immunized animals (rDSG3). * 0.05 < *p* < 0.01; ** *p* < 0.01. (**C**) Indirect immunofluorence on WT skin performed with serum of immunized animals. Scale bar: 50 μm.

**Figure 2 biology-12-00702-f002:**
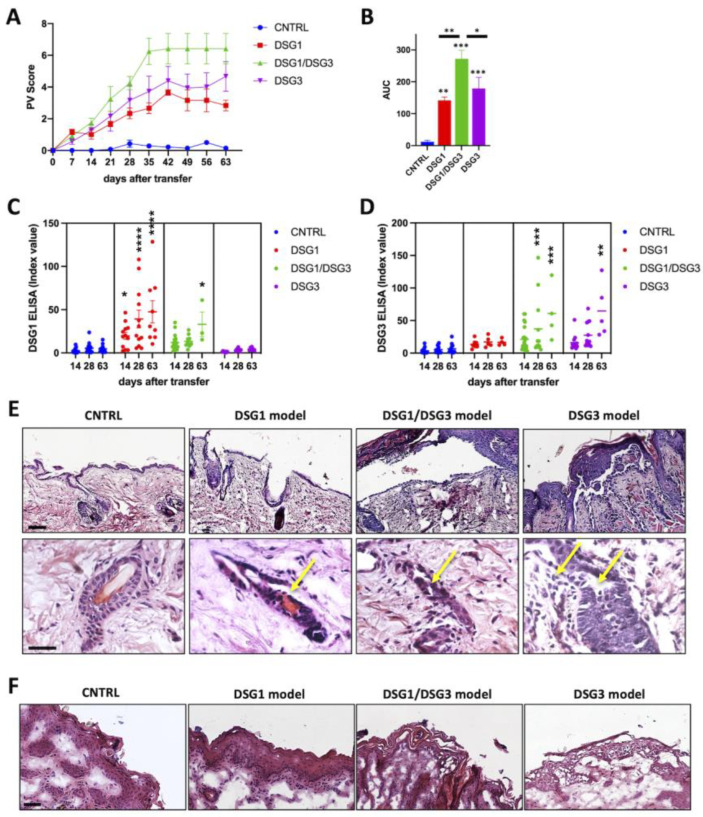
**Novel DSG1 and DSG1/DSG3 active pemphigus mouse models**. (**A**) Phenotypic aspects of animals receiving either DSG1, DSG1/DSG3, or DSG3 autoreactive splenocytes were described by PV score and evaluated weekly over a period of 63 days after splenocytes transfer into Rag2-/- mice. CNTRL animals received splenocytes from animals treated only with adjuvant. (n = 5–7 animals). Turkey’s multiple comparisons test and 2way-ANOVA show statistically significant differences among treated groups vs. CNTRL starting from 21 days, while among different models from day 35. In particular, day 21: ** DSG1/DSG3 vs. CNTRL. Day 28: *** DSG1/DSG3 vs. CNTRL; ** DSG3 vs. CNTRL. Day 35: **** DSG1/DSG3 vs. CNTRL; *** DSG3 vs. CNTRL; * DSG1/DSG3 vs. DSG1; * DSG1/DSG3 vs. DSG3. From day 42 till the end of the observational period: **** DSG1/DSG3 vs. CNTRL; **** DSG3 vs. CNTRL; * DSG1 vs. CNTRL; * DSG1/DSG3 vs. DSG1; * DSG1/DSG3 vs. DSG3. (**B**) PV score over time was also represented by an “Area Under the Curve” graphic (AUC). One-way ANOVA between treatments, *p* < 0.0001. Turkey’s multiple comparisons test were performed. All experimental groups are significantly different from CNTRL one, as shown. Moreover, DSG1/DSG3 is statistically different from both DSG1 and DSG3 alone. * 0.05 < *p* < 0.01; ** *p* < 0.01; *** *p* < 0.001 (**C**) DSG1 ELISA over time, 2way-ANOVA and Turkey’s multiple comparisons test were performed, and comparisons are made between control and treated groups at each timepoint. (**D**) DSG3 ELISA overtime, 2way-ANOVA and Turkey’s multiple comparisons test were performed, and comparisons are made between control and treated groups at each timepoint. (**E**) Representative pictures of the histologic examination of skin in active mouse models by H&E.; details of bulges are shown. Scale bar: 50 μm. (**F**) Representative pictures of the histologic examination of oral mucosa in active mouse models by H&E. Scale bar: 50 μm.

**Figure 3 biology-12-00702-f003:**
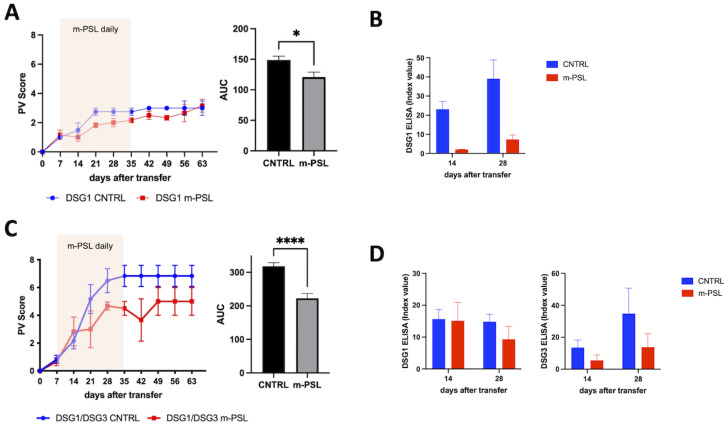
Effects of Methyl-Prednisolone (m-PSL) administration. (**A**) Effects in DSG1 pemphigus mouse model. m-PSL was given i.p. daily, starting at day 7 following the adoptive transfer, until day 35. Animals were allocated arbitrarily to the m-PSL or PBS (DSG1 CNTRL) treatment group (n = 3 animals per group). PV score was reported weekly until day 63. PV score over time was also shown as Area Under the Curve (AUC). (**B**) DSG1 ELISA over time. (**C**) Effect of m-PSL treatment in DSG1/DSG3 pemphigus mouse model. DSG1/DSG3 animals were treated as in A. * 0.05 < *p* < 0.01; **** *p* < 0.0001. (**D**) DSG1 and DSG3 ELISAs over time.

**Table 1 biology-12-00702-t001:** Type of lesions, sites of lesions and score for each symptom.

Type of Lesion	Score	Sites
Erosion	1	Snout
		Periocular region
		Periauricolar region
		Back
		Chest
		Abdomen
		Right foreleg
		Left foreleg
		Right hind leg
		Left hind leg
		Tail
Alopecia	1	Face
	0.5	Neck
	1	Back
	0.5	Abdomen
Erythema	0.5–1	Footpad
		Abdomen
		Face

## Data Availability

All data generated or analyzed during this study are included in this article. Further enquiries can be directed to the corresponding author.

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
