# Peer review of "A Novel In Vivo Active Pemphigus Model Targeting Desmoglein1 and Desmoglein3: A Tool Representing All Pemphigus Variants"

_biology, 2023, doi:10.3390/biology12050702_

Round 1

Reviewer 1 Report (Previous Reviewer 1)

Review to „A novel in vivo active pemphigus model targeting Desmoglein1 and Desmoglein3: a tool representing all Pemphigus variants“

Lotti et al. now present a revised version of a mixed Dsg1/Ds3 mouse model. I feel much as improved since the last version, and i still think that this mixed model is a good addition to the desmoglein / pemphigus community. I do however, have some minor issues

Minor points – Reference to Fig. 1C missing

-          Adjust layout of Fig. 1A/B to the other figures for conformity

-          L24 its „mouse model“ (if you talking singular)

-          Background – i have a hard time reading this paragraph. L32 atypical antigens is mentioned, again in l37 (unnecessary); l32 „with regards to …“!

-          L46 „Production of autoantibodies was evaluated by ELISA“. Unnessesary since no information in that sentence

-          L132 – (murine?) healthy skin slides

-          Finally, a manuscript with a detailed ethics statement (very good)

-          L228 „increase overtime of the antibody titers“ – in general the question, are there any statistics for Fig. 2A, C, D?

-          Is Dsg1/Dsg3 lower in titers than Dsg1 only (Fig. 2C)?

-          Figure 3 legend – rather start with „Effect of …

-          Fig. 3 adjust graphs from (A) and (B) to (C) and (D) for conformity

-          The discussion (and amount of references) feels very short (especially for a full article) since no data was actually was discussed here

minor editing necesseary

Author Response

We would like to thank the reviewer for comments and suggestions. Here a point by point answer to reviewer's comments (in general modifications of the manuscript text are highlighted in red):

- Reference to Fig. 1C missing 

R: Now we introduce it in lines 245 and 249.

- Adjust layout of Fig. 1A/B to the other figures for conformity 

R: Done!

- L24 its „mouse model“ (if you talking singular) 

R: Corrected

- Background – i have a hard time reading this paragraph. L32 atypical antigens is mentioned, again in l37 (unnecessary); l32 „with regards to …“!

R: We removed atypical.

- L46 „Production of autoantibodies was evaluated by ELISA“. Unnessesary since no information in that sentence

R: We changed the abstract accordingly

- L132 – (murine?) healthy skin slides

R: yes, murine healthy skin. Added!

- Finally, a manuscript with a detailed ethics statement (very good) -->

R: Thank you very much!

-          L228 „increase overtime of the antibody titers“ – in general the question, are there any statistics for Fig. 2A, C, D?

R: in Fig.2A  by 2way-ANOVA and Turkey's multiple comparisons there are statistically significant differences among treated groups vs CNTRL starting from 21 days, while among different models from day35. In particular, day 21: ** DSG1/DSG3 vs CNTRL. Day 28:  *** DSG1/DSG3 vs CNTRL; ** DSG3 vs CNTRL. Day 35:  **** DSG1/DSG3 vs CNTRL; *** DSG3 vs CNTRL; * DSG1/DSG3 vs DSG1; * DSG1/DSG3 vs DSG3. From day 42 till the end of the observational period:  **** DSG1/DSG3 vs CNTRL; **** DSG3 vs CNTRL; * DSG1 vs CNTRL; * DSG1/DSG3 vs DSG1; * DSG1/DSG3 vs DSG3. We added the description in the figure legend. For ELISAs, we added statical analysis (2way-ANOVA and Turkey's multiple comparisons) in the graphs: comparisons were made between control and treated groups at each timepoint.

 -          Is Dsg1/Dsg3 lower in titers than Dsg1 only (Fig. 2C)? 

R: Yes. Probably it depends on the fact that we transfer a decreased number of autoreactive splenocytes per each type in the mixed model.

-  Figure 3 legend – rather start with „Effect of …

R: Yes, thank you!

-Fig. 3 adjust graphs from (A) and (B) to (C) and (D) for conformity

R: Done! Corrected also in the figure legend.

-   The discussion (and amount of references) feels very short (especially for a full article) since no data was actually was discussed here

R: We enlarge the discussion paragraph and references list.

Reviewer 2 Report (Previous Reviewer 2)

Thank you for sharing the revised manuscript. I think the revised version is acceptable for publication in the Biology.

Reviewer 3 Report (New Reviewer)

Although the animal model for pemphigus vulgaris (PV) which is transferred autoreactive B/T cells specific to Dsg3 (mucosal dominant PV, in human) has been reported, the reviewer developed novel animal models which was transferred autoreactive T/B cells specific to both Dsg1 and Dsg3 (mucocutaneous PV) as well as specific to Dsg1 only (PF). The pathogenesis in pemphigus is heterogenous; 1) direct inhibition of steric hindrance by desmosomes targeting autoantibodies, 2) alterations of signaling pathways after the binding of antibodies, and 3) cellular immunity (especially in paraneoplastic pemphigus). Interestingly, activations of different signaling pathways by anti-Dsg1 and Dsg3 were reported (doi.org/10.3389/fmed.2021.701809).

These models allow us to evaluate the efficacy of a certain treatment in different autoimmunological reactions to each (Dsg1 or Dsg3) or combined antigens (Dsg1 and Dsg3). Moreover, these models may elucidate the different mechanism in vivo, especially in signaling pathway induced by autoantibodies targeting to Dsg1, Dsg3 or both Dsg1 and Dsg3.

1)     As the reviewer mentioned above, these animal models can be useful not only to evaluate the effectiveness of experimental therapies, but also to elucidate different activations of signaling pathway altered by different autoantibodies, including anti-Dsg1, anti-Dsg3 or anti-Dsg1/Dsg3. The authors can discuss on this possibility in the discussion.

2)     In Fig. 3A and 3B, the authors compared PV score in untreated- and treated-groups. Did the authors perform a statical comparison of PV score between each group at every time points (day7, 14, …, 63)?

3)     In Fig. 3, after the treatment, PV scores in each group got close. Did that occur resulting in reincrease of autoantibodies? The authors would be better to check autoantibodies titer with sera at days after the treatment (day 49 - 63), if sera are available.

4)     In PV scoring, why did the authors define the score for redness and erythema ranging from 0.5 to 1? How did distinguish 0.5 from 1.0? The scoring should be definitely.

Author Response

We would like to thank the reviewer for comments and suggestions. Here a point by point answer to reviewer's comments (in general modifications of the manuscript text are highlighted in red):

1)     As the reviewer mentioned above, these animal models can be useful not only to evaluate the effectiveness of experimental therapies, but also to elucidate different activations of signaling pathway altered by different autoantibodies, including anti-Dsg1, anti-Dsg3 or anti-Dsg1/Dsg3. The authors can discuss on this possibility in the discussion. 

R: Thank you for pointing out this very important topic. We added a paragraph in the discussion.

2)     In Fig. 3A and 3B, the authors compared PV score in untreated- and treated-groups. Did the authors perform a statical comparison of PV score between each group at every time points (day7, 14, …, 63)?

R: We presented statistical analysis only for the AUC graphs, in order to give general trend of the treatment. The low numbers of utilized animals did not allow us to have a robust analysis at each timepoint.

3)     In Fig. 3, after the treatment, PV scores in each group got close. Did that occur resulting in reincrease of autoantibodies? The authors would be better to check autoantibodies titer with sera at days after the treatment (day 49 - 63), if sera are available.

R: Thank you for the suggestion. Unfortunately, it is not possible, due to the fact that control animals very often died before the end of the experiment or they had to be sacrificed due to their severe conditions.

4)     In PV scoring, why did the authors define the score for redness and erythema ranging from 0.5 to 1? How did distinguish 0.5 from 1.0? The scoring should be definitely.

R: It depends on the extension and on the intensity of the redness.

This manuscript is a resubmission of an earlier submission. The following is a list of the peer review reports and author responses from that submission.

Round 1

Reviewer 1 Report

Dear editorial office, dear authors,

the present paper introduces a mixed Dsg1/Dsg3 driven transfer model of pemphigus vulgaris. Similar to (PMID: 31275323), proteins were used for immunization with subsequent transfer into Rag2-/- mice. While non-pathogenic antibodies were shown after immunization in another publication (PMID: 34731225), here a transfer presented in this paper is sufficient for acantholysis.

I do have several issues with the publication

1) english needs to be checked. Already the headline is wrong, the "model doesn't react"

2) why do you talk about atypical pemphigus models, this isn t part of the publication
3) Distinguishing active "transfer" models and active immunization models feels necessary, since there are mice (
PMID: 25252957) and models (for Dsc3 for instance, PMID: 34265330) that are not mentioned at all here. 
4) You introduced [9] as an active model against Atypical Pemphigus, yet it is a review which I also cannot connect to for some reason
5) M&M ELISA for Dsg1 and Dsg3 titers is missing
6) positive is the actual mentioning of the experimental murine registration number

7) results 3.1 is not results
--> if you claim that you can now dissect all pemphigus variants, i am missing in depth analysis of antibodies (keratinocyte assay e.g.) and target organs (comparison mucosa vs. skin)

Reviewer 2 Report

Manuscript ID biology-2190197 presents a novel in vivo active pemphigus model reacting against desmoglein (Dsg) 1 and 3. This is an interesting study trying to create a new mouse model of pemphigus foliaceus (PF) and mucocutaneous pemphigus vulgaris (MCPV). The authors should add some data to further develop the mouse model.

1. In Abstract, the authors should include more specific data like they show in Results.

2. In Materials and Methods, the authors should explain the details of the PV score evaluation in this manuscript. Otherwise, it would be hard for the readers to understand.

3. In Results, the authors need to show whether the immunized mice produced antibodies against Dsg1 and Dsg3, respectively.

4. When WT mice immunized with recombinant Dsg1 produced antibodies against Dsg1, the authors need to describe whether the immunized WT mice developed the phenotype of pemphigus. Also, the authors should show whether IgG deposition was observed in the skin of the immunized WT mice.

5. The authors should describe whether the mouse model of MCPV with antibodies to both Dsg1 and Dsg3 have mucosal lesions.

6.  Figure 1E is too small to determine if blisters are forming at the level of PF.

7. In Figure 2, the authors should show the changes in the serum antibody titers against Dsg1 and Dsg3 as well as PV scores.